# SGP-TOD: Building Task Bots Effortlessly via Schema-Guided LLM Prompting

**Xiaoying Zhang[1], Baolin Peng[2], Kun Li[1], Jingyan Zhou[1], Helen Meng[1,3]**

[1]The Chinese University of Hong Kong, Hong Kong
[2]Tencent AI Lab, Bellevue
[3]Centre for Perceptual and Interactive Intelligence, Hong Kong
{zhangxy, kunli, jyzhou, hmmeng}@se.cuhk.edu.hk
{baolinpeng}@global.tencent.com

## Abstract

Building and maintaining end-to-end task bots using minimal human effort is a long-standing challenge in dialog research. In this work, we introduce SGP-TOD, Schema-Guided Prompting for building Task-Oriented Dialog systems effortlessly based on large language models (LLMs). Utilizing the predefined task schema, *i.e.,* belief instruction and dialog policy, we instruct fixed LLMs to generate appropriate responses on novel tasks, without the need for training data. Specifically, SGP-TOD comprises three components: an LLM for interacting with users, a Dialog State Tracking (DST) Prompter to aid the LLM in tracking dialog states with the given belief instruction, and a Policy Prompter to direct the LLM to generate proper responses adhering to the provided dialog policy. Experimental results on Multiwoz, RADDLE, and STAR datasets show that our training-free strategy, SGP-TOD, yields state-of-the-art (SOTA) zero-shot performance, significantly surpassing the few-shot approaches. In a domain-extension setting, SGP-TOD aptly adapts to new functionalities by merely adding supplementary schema rules. We make our code and data publicly available.[1]

## 1 Introduction

Building task-oriented dialog (TOD) systems has been a long-standing challenge in artificial intelligence. The prevailing approach for creating task bots (Hosseini-Asl et al., 2020; Peng et al., 2021a; Sun et al., 2022) is to fine-tune pre-trained language models (PLMs), such as T5 (Raffel et al., 2020) and GPT-2 (Radford et al., 2019). Despite their great success, developing and maintaining such task bots generally requires adequate annotated data and extensive fine-tuning/re-training.

Recently, large Language Models (LLMs), such as ChatGPT (OpenAI, 2022) and GPT-4 (OpenAI,

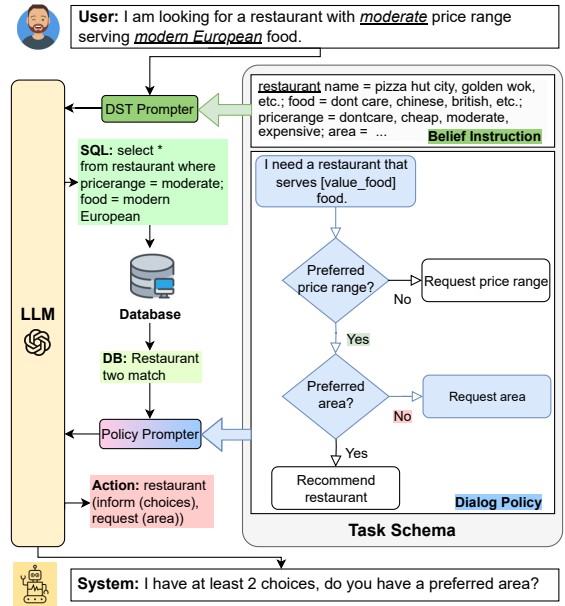

Figure 1: The proposed SGP-TOD is depicted with a dialog example, where the prompters integrate the task schema (right) to assist the frozen LLM in generating an appropriate response (left).

2023), have revolutionized natural language processing (NLP) applications (Wei et al., 2022; Wang et al., 2023), owing to their remarkable conversational skills (Qin et al., 2023), instruction-following abilities (Ouyang et al., 2022) and zero-shot generalization capabilities (Chowdhery et al., 2022a; Hu et al., 2022). This raises a research question: can LLMs be effectively utilized for building task bots with minimum human effort?

A contemporary study (Hudecek and Dusek, 2023) explores the potential of LLMs for rapidly building task bots via few-shot prompting, *a.k.a.* in-context learning (ICL) paradigm (Brown et al., 2020; Madotto et al., 2021). Though demonstrably effective, the ICL performance is highly influenced by the quality of the in-context exemplars (Zhao et al., 2021; Liu et al., 2022; Dong et al., 2023), as they struggle to provide comprehensive information for dialog task completion.

---

[1]https://github.com/zhangxy-2019/sgp-tod

In this work, we introduce symbolic knowledge (Nye et al., 2021; Cheng et al., 2023), *i.e.,* the task schema into LLMs, for creating task bots. Task schema (Mosig et al., 2020; Mehri and Eskenazi, 2021) encompasses a concise symbolic representation of a task, supplying LLMs with a comprehensive blueprint. It comprises (*i*) task-specific ontology containing all slots and their appropriate values (Budzianowski et al., 2018); and (*ii*) a dialog flow explicitly outlining fundamental interaction patterns (Peng et al., 2021b). Specifically, we propose SGP-TOD (as depicted in Figure 1), a schema-guided prompting method for rapidly building task bots. We integrate the predefined task schema and dialog context into prompts through the use of two specifically-designed prompters, namely a DST Prompter and a Policy Prompter. Utilizing these prompters, we adeptly guide fixed LLMs to track dialog states, retrieve database entries, and generate appropriate responses for novel tasks ***in a zero-shot manner, without the need for additional training or fine-tuning***. By incorporating task-specific symbolic knowledge into LLMs, SGP-TOD provides knowledge-based, coherent and human-like responses. Moreover, this training-free design empowers developers to flexibly prototype dialog systems on new tasks, while seamlessly extending system functionalities through modifying the task schema.

We perform empirical automatic evaluations on two multi-domain datasets, namely, Multiwoz 2.0 and 2.2 (Budzianowski et al., 2018; Zang et al., 2020), as well as two single-domain/task datasets, RADDLE (Peng et al., 2021a) and STAR (Mosig et al., 2020), within zero-shot scenarios. Additionally, we complement these assessments with interactive human evaluations. The results indicate that SGP-TOD, *employing merely task schema devoid of any training or fine-tuning*, substantially boosts the SOTA zero-shot results, markedly outperforming few-shot prompting/fine-tuning methods, and even attaining competitive results *cf.* full-shot fine-tuning approaches. In a domain-extension context, SGP-TOD proficiently adapts to new functionalities *by simply adding a handful of schema rules without necessitating further data collection*, significantly exceeding the few-shot prompting/fine-tuning methods reinforced by machine teaching (Williams and Liden, 2017).

In summary, our contributions are three-fold:

- We propose SGP-TOD, a schema-guided

LLM prompting strategy that facilitates in effortlessly creating task bots, eliminating the necessity for task-specific data or fine-tuning.

- We integrate symbolic knowledge – task schema into LLMs, allowing them to generate schema-compliant responses and adaptively expand their functionalities to tackle new tasks by solely modifying the task schema.

- We demonstrate the effectiveness of SGP-TOD on Multiwoz, RADDLE, STAR datasets in zero-shot settings using both automatic and human evaluations. SGP-TOD notably elevates the SOTA zero-shot performance.

## 2   Related work

**Zero-Shot Task-Oriented Dialog Modeling.** Zero-shot generalization is an essential yet challenging task in TOD research. A comprehensive study is shown in Appendix A. In this paper, we focus on zero-shot end-to-end dialog modeling, including policy management and dialog generation.

The works by Zhao and Eskenazi (2018) and Qian and Yu (2019) utilize ontology and response templates to train dialog models, enabling the discovery of shared dialog policies between the source and target domains. To enable broader adaptation to diverse dialog policies, Mosig et al. (2020); Mehri and Eskenazi (2021) implement task-specific policy skeletons, training dialog models to adhere to novel policies. Furthermore, Zhao et al. (2022) employs a neural language model (LM) for tracking dialog states and user actions using slot and action descriptions; subsequently, a policy program is deployed to facilitate an LM in generating system actions and responses. Despite the effectiveness of previous approaches, they still require ample fine-tuning and copious annotated dialog corpora on source or heterogeneous domains/tasks.

A concurrent study to ours is Hudecek and Dusek (2023), which employs a prompting strategy – IG-TOD (instruction-guided TOD) to guide frozen LLMs in generating suitable responses. Specifically, IG-TOD first tracks belief states by utilizing slot descriptions as prompts, then retrieves database entries, and generates responses. Our SGP-TOD differs in that: (*i*) we employ slot names and value examples, rather than slot descriptions, as prompts to facilitate frozen LLMs in generating belief states, thereby reducing human effort; (*ii*) we offer a policy skeleton to guide

LLMs in producing appropriate responses. In addition, experimental results indicate that SGP-TOD substantially outperforms IG-TOD.

**Leveraging LLMs for Dialog Tasks.** LLMs (Chowdhery et al., 2022b; OpenAI, 2023) have exhibited unparalleled mastery of natural language understanding, reasoning and generation (Wei et al., 2022; Bubeck et al., 2023). Three primary research directions have obtained substantial success in numerous dialog tasks by utilizing LLMs. (*i*) Few-shot prompting (Brown et al., 2020) has showcased remarkable performance in intent classification (Yu et al., 2021), semantic parsing (Shin and Van Durme, 2022), dialog state tracking (Hu et al., 2022; Xie et al., 2022), and response generation (Madotto et al., 2021). (*ii*) Li et al. (2022); Mehri et al. (2022); Dai et al. (2023) employ LLMs for data augmentation, *i.e.,* generating synthetic task-oriented dialogs to train smaller models for inference. (*iii*) Recently, several studies endeavor to support LLMs in specialized tasks by incorporating external knowledge. Peng et al. (2023) advocates for enhancing LLMs' responses with external knowledge and automated feedback to reduce hallucination. Liang et al. (2023) suggests connecting LLMs with millions of APIs to accomplish diverse tasks. Different from the aforementioned works, we aim to employ LLMs in building task bots in a zero-shot manner using pre-defined task schema.

# 3 SGP-TOD

## 3.1 Overview

The overall architecture of the proposed SGP-TOD (Figure 1) consists of three key components: (*i*) an **LLM**, responsible for adhering to instructions, comprehending user queries, and generating coherent responses for user interaction; (*ii*) a **DST Prompter**, tasked with supporting the LLM in tracking dialogue states using the belief instruction; (*iii*) a **Policy Prompter**, guiding the LLM to adhere to the predefined task policy for providing suitable system actions and responses.

At each dialog turn $t$, the end-to-end generation task is systematically divided into three subsequent sub-tasks: (*i*) **Belief State Prediction** – given the dialog history up to current dialog turn $h_t$, which is a sequence of utterances alternating between the user and the system $h_t = [u_1, r_1, u_2, r_2, \ldots, u_t]$ (where $u$ and $r$ denote user and system utterances, respectively), the DST Prompter embeds the belief instruction $\boldsymbol{BI}$ to direct the frozen LLM (pa-

rameterized by $\boldsymbol{\theta}$) in generating a belief state $\boldsymbol{b}_t$ (Equation 1). The belief state is then used to query a database and obtain the database (DB) state $\boldsymbol{c}_t$ (Equation 2). (*ii*) **System Action Determination** – the Policy Prompter incorporates a policy skeleton $\boldsymbol{PS}$, assisting the LLM in generating a system action $\boldsymbol{a}_t$, based on $\boldsymbol{h}_t$, $\boldsymbol{b}_t$, and $\boldsymbol{c}_t$ (Equation 3). (*iii*) **Dialog Response Generation** – grounded in the dialog history $\boldsymbol{h}_t$, belief state $\boldsymbol{b}_t$, DB state $\boldsymbol{c}_t$, system action $\boldsymbol{a}_t$, the Policy Prompter aids the LLM in generating a delexicalized response by providing the policy skeleton $\boldsymbol{PS}$ (Equation 4). Ultimately, the delexicalized response is automatically post-processed to generate system response in natural language. Detailed illustration with a dialog example is shown in Appendix L.

$$\boldsymbol{b}_t = LLM_{\boldsymbol{\theta}}(\boldsymbol{h}_t, \boldsymbol{BI}) \tag{1}$$
$$\boldsymbol{c}_t = DB(\boldsymbol{b}_t) \tag{2}$$
$$\boldsymbol{a}_t = LLM_{\boldsymbol{\theta}}(\boldsymbol{h}_t, \boldsymbol{b}_t, \boldsymbol{c}_t, \boldsymbol{PS}) \tag{3}$$
$$\boldsymbol{r}_t = LLM_{\boldsymbol{\theta}}(\boldsymbol{h}_t, \boldsymbol{b}_t, \boldsymbol{c}_t, \boldsymbol{a}_t, \boldsymbol{PS}) \tag{4}$$

## 3.2 LLM

An LLM is responsible for following task-specific instructions and generating appropriate responses.

Many off-the-shelf LLMs, *e.g.,* ChatGPT, Codex (Chen et al., 2021), are pre-trained on massive corpora of text data and/or code data. In addition, they are trained to follow instructions in the prompts (Ouyang et al., 2022) and provide pertinent responses. Exhibiting remarkable proficiencies in natural language processing, instruction compliance, and zero-shot generalization across diverse downstream dialog tasks, these LLMs serve as valuable foundation models for our approach.

## 3.3 DST Prompter

Given the dialog history $\boldsymbol{h}_t$, the DST prompter aims to guide the LLM in predicting the belief state $\boldsymbol{b}_t$ at each turn $t$, using the belief instruction $\boldsymbol{BI}$. The belief state $\boldsymbol{b}_t$ is defined as the concatenation of the domain/task (*i.e.,* user intent) $\boldsymbol{d}_t$ and a set of slot-value pairs $\left\{(\boldsymbol{s}_t^i, \boldsymbol{v}_t^i); i = 1, \ldots, n_t\right\}$, where $n_t$ is the total number of pairs in the set.

As shown in Figure 2, the proposed DST prompter contains four parts: (*i*) a *task instruction* that offers general guidance on belief state prediction;[2] (*ii*) *belief instructions* $\boldsymbol{BI}$ of all domains/tasks; (*iii*) a *formatting example* illustrating

---

[2] We assess several task instructions written by different authors, yielding minor performance disparities.

**DST Prompter**

**Task instruction**

Following the instructions, predict the belief state based on the history.

**Belief instructions**

restaurant name = pizza hut city, golden wok, etc.; food = dont care, chinese, mediterranean, british, etc.; pricerange = dontcare, cheap, moderate, expensive; area = dont care, centre, east, north, south, west ; booking_day = monday, tuesday, wednesday, thursday, friday, saturday, sunday; booking_people = 1,2,3,4,5,6,7; booking_time = 12:15, 13:30, etc.

attraction type = swimmingpool, theatre, architecture, museum, nightclub, boat, park, college, concerthall, entertainment, multiple sports, cinema; area = dont care, centre, east, north, south, west; ....

**Formatting example**

**history**

user: Would you be able to tell me whether there are any *multiple sports* venues in the *center* of town?

SQL: select * from attraction where type = multiple sports; area = centre

**Test (on target task/domain)**

**history**

user: I am looking for a restaurant in the *moderate* price range serving *modern European* food. system: I have at least 2 choices, do you have a preferred area in mind? user: I'd like a restaurant on the *south* end of town, please.

**LLM** 🌐

SQL: select * from restaurant where pricerange = moderate; food = modern European; area = south

Figure 2: Illustration of belief state prediction utilizing DST Prompter. The predicted belief state is highlighted.

the anticipated output format to direct the LLM, in addition, we follow Hu et al. (2022) and adopt SQL state to represent the dialog state $b_t$[3]; and (*iv*) the *test input*, *i.e.,* the given dialog history $h_t$. Since the prompt is fixed and no labeled data from the target task or domain is used, we refer to this setting as "zero-shot", following Wang et al. (2022b).

**Belief Instruction.** For each task/domain, the belief instruction contains the task/domain name, all potential slot names, and their possible values (Figure 2). Regarding categorical slots, such as the "price range" in the restaurant domain, all plausible values are included, *i.e.,* "don't care", "cheap", "moderate", and "expensive"; whereas, for non-categorical slots, such as "name", only a few value examples are injected, *e.g.,* Pizza Hut City, Golden Wok, etc.[4] Detailed belief instructions for all tasks/domains can be found in Appendix B.

---

[3]SQL: select * from $d_t$ where $s_t^1 = v_t^1; \dots; s_t^{n_t} = v_t^{n_t}$.

[4]We assess belief instructions with diverse slot value examples, revealing minor performance variations.

**Policy Prompter**

**Task instruction**

Following the instructions, generate appropriate response based on the history.

**Formatting example (from other task/domain)**

...

**Policy skeleton (on target task/domain)**

(1) **user:** I'm looking for a restaurant that offers [value_food] food in a [value_pricerange] price range. **action:** restaurant (inform (choices), require (area)) **system:** I have over [value_count] restaurant -s to choose from, do you have a preferred area in mind? [eos]

(2) **user:** I need a restaurant that serves [value_food] food in a [value_pricerange] on the [value_area] side of town. **action:** restaurant (recommend (name), inform (food, pricerange, area)) **system:** how about [restaurant_name]? It serves [value_food] food in the [value_pricerange] price range in the [value_area]. [eos] ...

(17) **DB:** restaurant one match. **action:** restaurant (recommend (name)) **system:** How does [restaurant_name] sound?

**Test (on target task/domain)**

**history**

user: I am looking for a restaurant in the *moderate* price range serving *modern European* food. system: I have at least 2 choices, do you have a preferred area in mind? user: I'd like a restaurant on the *south* end of town, please.

SQL: select * from restaurant where pricerange = moderate; food = modern European; area = south DB: Restaurant one match

**LLM** 🌐

action: restaurant (recommend (name), inform (food, pricerange, area)) system: How does [restaurant_name] sound? It serves [value_food] food in the [value_pricerange] price range on the [value_area] side of town. [eos]

Figure 3: Illustration of system action determination and response generation employing the Policy Prompter. The pertinent template turns , previously predicted belief state , retrieved DB state within the input, alongside the generated system action and generated response in the output are accentuated.

### 3.4 Policy Prompter

Dialog policy, governing the behavior of task bots, plays a crucial role in task-oriented dialogs. To represent the dialog policy for a given task, we utilize a *policy skeleton*, which delineates interaction patterns and encompasses business logic in the form of template dialog flows (Peng et al., 2021b). The Policy Prompter is devised to guide the static LLM in adhering to the policy skeleton $PS$, enabling the sequential generation of appropriate system actions $a_t$ and responses $r_t$.

Analogous to the DST Prompter, the Policy Prompter (Figure 3) comprises four components: (*i*) a *task instruction*; (*ii*) a *formatting example* derived from another task/domain, consisting of a par-

tial policy skeleton and its associated dialogue turn exemplar (in Appendix C); (*iii*) a *policy skeleton* for the previously predicted domain/task; and (*iv*) the *test input*, *i.e.,* the dialog history $h_t$, generated belief state $b_t$, and obtained DB state $c_t$.

**Policy Skeleton.** Given that user behaviors and DB results jointly determine system actions and responses, policy skeleton is designed to cover all fundamental user behaviors and characteristic DB results, along with their corresponding system actions and responses.[5] Considering the infeasibility of developing a multi-task/domain policy skeleton for every possible combination of tasks and domains, we opt to develop a distinct policy skeleton tailored to each specific task and domain.

Following Mehri and Eskenazi (2021), our strategy converts the established dialog policy into a series of template dialog turns $\mathcal{X}$ that are logically arranged and concentrate on task completion:

$$\begin{aligned} \mathcal{X} &= \{\boldsymbol{x}_i\}_{i=1}^N, \\ \boldsymbol{x}_i &= (u^i, a^i, r^i) or (c^i, a^i, r^i) \end{aligned} \quad (5)$$

where $\boldsymbol{x}_i$ is a template dialog turn, which contains a user utterance $u^i$ or a DB state $c^i$, matching system action $a^i$, and system response $r^i$. $N$ denotes the total number of template turns within the policy skeleton (around 10-20 template turns depending on the task complexity). In order to equip the frozen LLM with new capabilities or modify current ones, we only need insert, amend, or eliminate a few template turns within the policy skeleton.

## 4 Experiments

### 4.1 Experimental Setup

**Datasets.** (*i*) Two **multi-domain** dialog datasets: `Multiwoz 2.0` (Budzianowski et al., 2018) and `Multiwoz 2.2` (Zang et al., 2020). (*ii*) Two **single-domain/task** datasets: `RADDLE` (Peng et al., 2021a,c) and `STAR` (Mosig et al., 2020) (single-task dialogs from the corpus, following the "happy path"). Details are elaborated in Appendix D.

**Automatic Evaluation Metrics.** We evaluate the end-to-end dialog generation performance using the same metrics as those listed in Budzianowski et al. (2018): `Inform`(%), `Success`(%), `BLEU`(%) (Papineni et al., 2002) and `Combined`(%) judges the overall quality, defined as `Combined` =

---

[5]We do not enumerate every conceivable combination of user behaviors or potential database results, as schema engineering is not the primary focus of this study.

---

(`Inform` + `Success`) $\times$ 0.5 + `BLEU`. Additionally, we utilize `BERTScore`(%) (Zhang* et al., 2020).

Following Mehri and Eskenazi (2021), we perform the next action prediction task on `STAR` (wherein the system actions and response templates are mapped one to one), which predicts next system action given the dialog history. We report the results using `F1score`(%) and `accuracy`(%).

**Human Evaluation Metrics.** We conduct interactive human evaluations (by five student helpers), following the evaluation protocol in the DSTC9 Track 1 challenge (Gunasekara et al., 2020). For each dialog session, students are mandated to interact with a dialog agent via natural language and assess the overall dialog quality employing these five metrics: (*i*) `Success w/o g`(%), (*ii*) `Success w/ g`(%), (*iii*) `Understanding`(1-5), (*iv*) `Appropriateness`(1-5) and (*v*) `Turns`. Full details are elaborated in Appendix D.

**Compared Methods.** We compare the proposed SGP-TOD with SOTA zero-shot transfer methods and zero-shot/few-shot prompting strategies. (We report the mean results of three different runs.)

**Zero-shot transfer methods:**

- BERT+S (Mosig et al., 2020) augments a BERT-base classifier (Devlin et al., 2019) with a system-side schema to predict system action.

- SAM (Mehri and Eskenazi, 2021) is based on BERT-base, which uses a user-aware schema to predict the next system action.

- ANYTOD-XXL (Zhao et al., 2022) adopts T5-XXL (Roberts et al., 2022) to generate system actions and responses utilizing slot/action descriptions and a policy program. It is pretrained on SGD dataset (Rastogi et al., 2020a).

**Prompting methods:**

- IG-TOD-CHATGPT (Hudecek and Dusek, 2023) is a prompting approach based on ChatGPT, exploiting slot descriptions for tracking dialog states, fetching DB entries, and generating responses. IG-TOD-CHATGPT-ZS and IG-TOD-CHATGPT-FS are in the zero-shot and few-shot settings, respectively.

- FEW-SHOT-CHATGPT is a few-shot prompting approach applied to ChatGPT, utilizing a few (*i.e.,* $k$) training dialog turns as prompts. Optimal results are achieved with $k = 15$ on `Multiwoz` and $k = 10$ on `RADDLE`.

- SGP-TOD (Ours) is compatible with any off-the-shelf LLMs. In this paper, we employ

| Model | Multiwoz 2.0 | | | | Multiwoz 2.2 | | | |
|---|---|---|---|---|---|---|---|---|
| | Inform | Success | BLEU | Combined | Inform | Success | BLEU | Combined |
| *Full-shot fine-tuning (with 8.4k+ training dialogs):* | | | | | | | | |
| DAMD (Zhang et al., 2020) | 76.33 | 60.40 | 16.60 | 84.97 | - | - | - | - |
| SimpleTOD (Hosseini-Asl et al., 2020) | 84.40 | 70.10 | 15.01 | 92.26 | - | - | - | - |
| Soloist (Peng et al., 2021a) | 85.50 | 72.90 | 16.54 | 95.74 | 81.70 | 67.10 | 13.60 | 88.00 |
| PPTOD (Su et al., 2022) | 89.20 | 79.40 | 18.62 | 102.92 | - | - | - | - |
| Mars (Sun et al., 2022) | 88.90 | 78.00 | 19.90 | 103.35 | 88.90 | 78.00 | 19.60 | 103.05 |
| *Zero-shot transfer method (pre-trained on SGD):* | | | | | | | | |
| AnyTOD-XXL | - | - | - | - | 73.90 | 24.40 | 3.40 | 52.55 |
| *Few-shot prompting:* | | | | | | | | |
| IG-TOD-ChatGPT-fs | - | - | - | - | - | 20.00 | 7.17 | - |
| Few-Shot-ChatGPT | 44.74 | 24.32 | 7.88 | 42.41 | 45.40 | 24.50 | 7.72 | 42.67 |
| *Zero-shot prompting:* | | | | | | | | |
| IG-TOD-ChatGPT-zs | - | - | - | - | - | 15.00 | 3.58 | - |
| SGP-TOD-ChatGPT (Ours) | 64.56 | 54.05 | 7.17 | 66.48 | 64.70 | 54.70 | 6.96 | 66.66 |
| SGP-TOD-Codex (Ours) | 71.67 | 52.55 | 7.91 | 70.02 | 75.50 | 52.30 | 6.62 | 70.53 |
| SGP-TOD-GPT3.5 (Ours) | **83.88** | **69.87** | **9.09** | **85.97** | **82.00** | **72.50** | **9.22** | **86.47** |

Table 1: End-to-end dialog generation evaluation results on Multiwoz. Results of Soloist, Mars, AnyTOD-XXL on Multiwoz 2.2 are cited from Zhao et al. (2022). Results of IG-TOD-ChatGPT are cited from Hudecek and Dusek (2023). Other results of the full-shot fine-tuning methods are cited from He et al. (2022) and Sun et al. (2022). (Difference in mean is significant with p<0.01.)

| Model | Attraction | | | | Train | | | | Hotel | | | | Restaurant | | | |
|---|---|---|---|---|---|---|---|---|---|---|---|---|---|---|---|---|
| | Info. | Succ. | BLEU | Comb. | Info. | Succ. | BLEU | Comb. | Info. | Succ. | BLEU | Comb. | Info. | Succ. | BLEU | Comb. |
| *Few-shot fine-tuning (with 50 training dialogs):* | | | | | | | | | | | | | | | | |
| SimpleTOD | 65.66 | 46.97 | 5.85 | 62.17 | 59.00 | 44.00 | 7.07 | 58.57 | 62.50 | 40.00 | 7.70 | 58.95 | 75.50 | 44.50 | 11.00 | 71.00 |
| Soloist | 86.00 | 65.00 | **12.90** | 88.40 | 80.81 | 64.65 | **9.96** | 82.69 | 74.50 | 43.50 | **8.12** | 67.12 | 81.00 | 55.50 | 12.80 | 81.50 |
| *Few-shot prompting:* | | | | | | | | | | | | | | | | |
| Few-Shot-ChatGPT | 75.00 | 67.00 | 8.22 | 79.23 | 79.80 | 65.66 | 8.12 | 80.85 | 51.00 | 26.50 | 5.80 | 44.55 | 80.00 | 55.50 | 7.71 | 75.46 |
| *Zero-shot prompting:* | | | | | | | | | | | | | | | | |
| SGP-TOD-ChatGPT | 95.00 | **94.00** | 7.13 | 101.63 | 76.77 | 74.24 | 6.75 | 82.26 | 76.50 | 57.00 | 5.16 | 71.91 | 90.00 | 82.50 | 6.72 | 92.97 |
| SGP-TOD-Codex | **98.00** | 93.00 | 10.45 | **105.95** | 78.79 | 70.20 | 8.56 | 83.06 | **83.50** | 69.50 | 7.86 | **84.36** | 91.00 | **85.00** | 10.50 | 98.50 |
| SGP-TOD-GPT3.5 | 96.00 | 93.00 | 9.53 | 104.03 | **82.83** | **77.27** | 8.72 | **88.77** | 82.50 | 71.50 | 7.05 | 84.05 | **91.50** | 84.00 | **12.90** | **100.65** |

Table 2: End-to-end dialog generation evaluation results on RADDLE. The few-shot fine-tuning results are cited from Peng et al. (2021a). (Difference in mean is significant with p<0.01.)

ChatGPT, GPT-3.5 and Codex. Implementation details are provided in Appendix E.

## 4.2 End-to-End Evaluation on Multiwoz

**Results.** We present the evaluation results in multi-domain contexts on Multiwoz in Table 1. In addition to the aforementioned methods, we include the results of SOTA full-shot fine-tuning approaches to facilitate a more comprehensive comparison. SGP-TOD obtains SOTA *zero-shot performance*, substantially outperforming few-shot prompting approaches across all metrics, while even exhibiting competitive results in comparison to full-shot fine-tuning methods concerning Success and Inform. This confirms the effectiveness of integrating the task schema with the LLMs' proficient language processing capabilities.

***Comparison with Prompting Methods.*** SGP-TOD-ChatGPT distinctly surpasses the zero-shot prompting approach IG-TOD-ChatGPT-zs with respect to Success (surpassing by $40\%$) and BLEU (exceeding by $3\%$). Moreover, SGP-TOD-ChatGPT, *without requiring task-specific data,*

considerably outperforms the few-shot prompting methods, *i.e.,* IG-TOD-ChatGPT-fs and Few-Shot-ChatGPT (*e.g.,* about 30 points improvement over Success). This suggests that providing explicit and concise task instructions via task schema is preferable to imparting implicit task guidance through the selected dialog turns.

***Comparison with Zero-Shot Transfer Methods.*** Our SGP-TOD demonstrates a substantial advantage over AnyTOD-XXL, which necessitates task-specific pre-training and additional annotations, *e.g.,* slot and action descriptions, over all the metrics. This exemplifies the potency of SGP-TOD, which markedly reduces the necessity for human labor and computational resources.

***Comparison with Full-Shot Fine-Tuning Methods.*** SGP-TOD exhibits competitive performance over Inform and Success. The lower BLEU is due to a lack of linguistic variations of the template utterances, which is acceptable considering the trade-off between human effort and efficacy.

| Model | Task transfer | | Domain transfer | |
|---|---|---|---|---|
| | F1 | Accuracy | F1 | Accuracy |
| *Zero-shot transfer* | | | | |
| *(leave-one fune-tuning with 2.5k training dialogs):* | | | | |
| BERT+S | 24.25 | 24.89 | 25.70 | 28.56 |
| SAM | 49.82 | **51.30** | **55.91** | **57.92** |
| *Zero-shot prompting:* | | | | |
| SGP-TOD-CODEX-INI | 45.18 | 47.99 | 47.21 | 49.97 |
| SGP-TOD-GPT3.5 | 47.67 | 48.27 | 49.76 | 50.39 |
| SGP-TOD-CODEX | 49.78 | 51.01 | 52.72 | 53.66 |
| SGP-TOD-GPT3.5-E2E | **50.84** | 50.74 | 53.50 | 53.21 |

Table 3: Zero-shot end-to-end next action prediction evaluation results on STAR. (Difference in mean is significant with p<0.01.)

## 4.3 End-to-End Evaluation on RADDLE

**Results.** Table 2 reports the results in single-domain settings on RADDLE. On all four dialog tasks, SGP-TOD demonstrates remarkable zero-shot performance that consistently surpasses both few-shot prompting and fine-tuning approaches. This results in substantial improvements of up to 12% in Inform, 45% in Success, and 19% in Combined metrics, while maintaining competitive BLEU scores. This evidence further substantiates the efficacy of SGP-TOD.

## 4.4 End-to-End Evaluation on STAR

**Setup.** BERT+S, SAM are fine-tuned on source tasks/domains then zero-shot on the held-out task/domain.[6] SGP-TOD is presented with two formatting turns from the source tasks/domains.

**Results.** Following Mehri and Eskenazi (2021), we report the zero-shot evaluation results in two settings, *i.e.,* task transfer and domain transfer in Table 3. SGP-TOD, *merely with two formatting sample turns*, demonstrates exceptional performance, surpassing or rivaling SOTA zero-shot transfer methods in both settings. This outcome signifies that, even when faced with complicated business logic and system actions in dialog policies, the proposed SGP-TOD continues to exhibit commendable performance. Additionally, we investigate the impact of changing the number of training dialogs and formatting example turns in Appendix F.

***Impact of Different LLMs and Prompting Formats.*** SGP-TOD-CODEX surpasses SGP-TOD-GPT3.5 while rivaling SGP-TOD-GPT3.5-E2E (with template responses affixed to action labels in the policy prompt, demonstrated in Figure 10 in Appendix M). We conjecture that Codex, benefiting from extensive pre-training on copious code

---

[6]ANYTOD-XXL requires additional annotations, *e.g.,* belief descriptions, which makes it not suitable for STAR.

| Model | FT/FS/ZS | Restaurant-Ext | | | |
|---|---|---|---|---|---|
| | | Info. | Succ. | BLEU | BERTS. |
| *Without domain-relevant knowledge* | | | | | |
| ChatGPT | ZS | 44.00 | 6.00 | 4.31 | 85.96 |
| GPT-3.5 | ZS | 34.00 | 16.00 | 8.70 | 84.31 |
| *With prior knowledge on Restaurant* | | | | | |
| SOLOIST | FT | 78.00 | 0.00 | 10.62 | 87.24 |
| SGP-TOD-CHATGPT | ZS | 88.00 | 34.00 | 5.45 | 86.11 |
| SGP-TOD-GPT3.5 | ZS | 94.00 | 30.00 | 10.68 | 87.30 |
| *With knowledge on Restaurant-Ext* | | | | | |
| SOLOIST+TEACH | FT | 82.00 | 38.00 | 10.99 | 87.66 |
| FEW-SHOT-GPT3.5+TEACH | FS | 88.00 | 54.00 | 12.95 | 88.90 |
| SGP-TOD-CHATGPT-EXT | ZS | 88.00 | 78.00 | 6.25 | 86.15 |
| SGP-TOD-GPT3.5-EXT | ZS | **96.00** | **86.00** | **14.57** | **89.01** |

Table 4: End-to-end evaluation results on domain extension. FT: fine-tuning, FS: few-shot prompting, ZS: zero-shot prompting, Info.: Inform, Succ.: Success, BERTS.: BERTScore. (Difference in mean is significant with p<0.01.)

data, demonstrates enhanced proficiency compared to GPT-3.5 in interpreting action labels. In addition, appending template responses is presumed to facilitate the explication of action labels for GPT-3.5.

***Impact of Different Task Schemas.*** SGP-TOD-CODEX-INI, utilizing an identical task schema as employed in training SAM, manifests commendable performance. This result highlights that SGP-TOD as a flexible prompting strategy, compatible with any manually-crafted task schema.

## 4.5 End-to-End Evaluation on Domain Extension

**Setup.** We conduct experiments in a domain extension setting (Gasic et al., 2014; Lipton et al., 2018) to assess the efficacy of SGP-TOD in adapting deployed task bots to incorporate novel functionalities. Following Zhang et al. (2022), we construct the Restaurant-ext corpus by extending the Restaurant in RADDLE with four new slots: *[restaurant_dish]*, *[value_price]*, *[start_time]*, and *[end_time]*. Details are shown in Appendix J.

**Compared Methods.**

- ChatGPT, GPT-3.5 denote zero-shot prompting that receive two formatting examples.

- SGP-TOD-CHATGPT, SGP-TOD-GPT3.5 represent our SGP-TOD implementation, with the Restaurant policy skeleton.

- SOLOIST is trained with 50 training dialogs in Restaurant domain (reported in Table 2).

- SOLOIST+TEACH is fine-tuning method enhanced with machine teaching (Simard et al., 2017). We deploy SOLOIST to converse with

real users, then implement machine teaching to obtain 10/50/50 annotated dialogs in `Restaurant-ext` for training, validating, and testing. We fine-tune SOLOIST with the gathered 10 training dialogs covering new slots.

- FEW-SHOT-GPT3.5+TEACH is the few-shot prompting strategy augmented with machine teaching. We use 10 randomly selected dialog turns from the collected 10 training dialogs as prompts (with peak performance at 10).

- SGP-TOD-CHATGPT-EXT, SGP-TOD-GP3.5-EXT refer to SGP-TOD with `Restaurant-Ext` policy skeleton, where we only add four template turns about four new slots to the policy skeleton of `Restaurant`.

**Results.** In Table 4, SGP-TOD-CHATGPT-EXT, and notably SGP-TOD-GPT3.5-EXT surpasses all other evaluated approaches by a substantial margin over all the metrics. This demonstrates the strong adaptability of our SGP-TOD in accommodating novel functionalities, revealing its immense potential for lifelong learning. Two interactive dialog examples are supplied in Appendix K.

***Comparison with Approaches Augmented by Machine Teaching.*** SOLOIST yields zero `Success`, a predictable result given its lack of awareness regarding the new features. Augmented by machine teaching, SOLOIST+TEACH substantially improves SOLOIST in terms of `Inform` and `Success`. Nevertheless, relying solely on prior `Restaurant` knowledge, both SGP-TOD-CHATGPT and SGP-TOD-GP3.5 exhibit performance on par with SOLOIST+TEACH, demonstrating that SGP-TOD provides enhanced robustness in zero-shot generalization. Moreover, SGP-TOD-GP3.5-EXT obtains substantially higher `Success` rates than SOLOIST+TEACH (a rise of $48\%$) and FEW-SHOT-GPT3.5+TEACH (an increase of $32\%$). Compared to fine-tuning/prompting strategies utilizing additional dialogs corrected through machine teaching, SGP-TOD facilitates a more agile adaptation to novel functionalities by merely modifying template turns within the task schema.

## 4.6 Interactive Human Evaluation

**Setup.** We conduct interactive human evaluations on `Restaurant` domain to evaluate the performance of SOLOIST, FEW-SHOT-CHATGPT, SGP-TOD-CHATGPT (reported in Table 2), with 50 dialogs gathered for analysis, respectively. Details can be found in Appendix H.

| Model | Restaurant | | | | |
|---|---|---|---|---|---|
| | S w/o g ↑ | S w/ g ↑ | Und. ↑ | App. ↑ | T. ↓ |
| SOLOIST | 34.00 | 30.00 | 2.18 | 2.10 | 10.64 |
| FEW-SHOT-CHATGPT | 94.00 | 74.00 | 4.58 | 4.72 | 8.32 |
| SGP-TOD-CHATGPT | **100.00** | **92.00** | **4.86** | **4.88** | **7.28** |

Table 5: Human evaluation results. S w/o g, S w/ g: Success without / with grounding; Und.: Understanding; App.: Appropriateness; T.: Turns.

| Model | Multiwoz 2.2 | | | |
|---|---|---|---|---|
| | Inform | Success | BLEU | Combined |
| SP-TOD-GPT3.5 | **82.00** | **72.50** | **9.22** | **86.47** |
| -policy | 81.80 | 56.20 | 6.63 | 75.63 |
| -policy -DB | 81.40 | 52.30 | 6.57 | 73.42 |
| -policy -DB -belief | 38.60 | 33.90 | 6.29 | 42.54 |

Table 6: Ablation study results on the impact of the three components in the proposed SGP-TOD and the database expertise on `Multiwoz 2.2` using GPT-3.5. -policy: removing Policy Prompter, -DB: removing database expertise, -belief: removing DST Prompter.

**Results.** In Table 5, our proposed SGP-TOD-CHATGPT attains a remarkably high performance in a zero-shot context, consistently outpacing SOLOIST and FEW-SHOT-CHATGPT across all metrics. Particularly, regarding `Success w/ g`, SGP-TOD-CHATGPT significantly surpasses FEW-SHOT-CHATGPT (by $18\%$) and SOLOIST (by $62\%$), illustrating its proficiency in accomplishing tasks within real-world scenarios. Furthermore, SGP-TOD-CHATGPT exhibits a more stable performance (demonstrated in Appendix I). A detailed analysis is provided in Appendix I.

## 4.7 Ablation Study

In Table 6, we study the impact of the three components of SGP-TOD (namely, Policy Prompter, DST Prompter, and LLM) as well as the database expertise, on `Multiwoz 2.2` utlizing GPT-3.5. Combining the three elements in SGP-TOD with the database expertise produces the optimal result, underscoring the value of enhancing the LLM with the task schema and external database information. Detailed analyses are provided in Appendix G.

## 5 Conclusion

We present SGP-TOD, a schema-guided prompting strategy aimed at the expeditious construction of end-to-end task bots, relying exclusively on LLMs and the corresponding task schema. Employing the symbolic knowledge – task schema, SGP-TOD guides fixed LLMs to generate suitable responses for novel tasks in a zero-shot fashion. Empirical findings on four well-studied datasets

reveal that SGP-TOD attains remarkable SOTA zero-shot performance, using both automatic and human evaluations. For future work, we plan to explore the use of SGP-TOD to develop personalized chatbots by utilizing pertinent task schema.

## Limitations

This work is accompanied by two primary limitations. (*i*) Data contamination (Brown et al., 2020; Madotto et al., 2021) in prompt-based zero-shot learning pertains to the potential presence of test samples during the LLM pre-training phase. Given that data utilized for pre-training LLMs, such as ChatGPT and GPT-4, remains undisclosed and continuously expands, verifying data contamination presents a formidable challenge. Consequently, our research cannot preclude data contamination in the experimental process, deferring a more comprehensive investigation to future endeavors. Nevertheless, we undertake domain-extension experiments (Table 4 in Section 4.5), subjecting our proposed SGP-TOD to evaluation on a novel test set (currently not publicly available), encompassing recently obtained and annotated human-bot dialogs. The remarkable zero-shot performance of SGP-TOD demonstrates its substantial potential for adeptly adapting to innovative functionalities, without reliance on task-specific data.

(*ii*) We employ the manually-crafted task schema as prompts to steer the LLMs towards generating suitable responses on novel tasks. As illustrated in Table 3 of Section 4.4, SGP-TOD exhibits minor performance discrepancies when implementing disparate task schema formulated by various authors. Notwithstanding such variations, our objective is to offer a foundational basis for schema-guided LLM prompting; future research may investigate approaches to designing more efficient task schema, *i.e.,* diverse formats and coverage.

## Ethics Statement

Throughout the interactive human evaluations and domain-extension experiments, all participating student helpers were informed of the research objectives prior to the collection and annotation of human-bot dialog logs. Their privacy was ensured to remain protected and undisclosed during the research period. Each participant received equitable remuneration.

The Prompters utilized in this research incorporate no language that discriminates against specific individuals or groups (Zhou et al., 2022) and avoid any negative impact on users' well-being (Bergman et al., 2022). Instances of these Prompters are provided in Appendix M. Furthermore, subsequent research endeavors may consider utilizing the OpenAI moderation API[7] in conjunction with other related APIs to systematically filter out unsuitable user inputs and system responses.

## Acknowledgements

This Project is partially supported by the HKSARG General Research Fund (Ref No. 14207619). We would like to express our gratitude to Xiaohan Feng and Haohan Guo for their valuable comments.

---

[7]https://platform.openai.com/docs/guides/moderation/overview

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

Figure 4: Detailed belief instructions in DST Prompter.

## A  Zero-Shot Task-Oriented Dialog Modeling.

Table 7 summarizes four main research directions in zero-shot task-oriented dialog modeling: slot filling (SF), dialog state tracking (DST), end-to-end policy management (E2E policy) and end-to-end dialog generation (E2E dialog).

## B  Detailed Belief Instructions in DST Prompter

Figure 4 shows the detailed belief instructions in DST Prompter.

## C  A Formatting Example in Policy Prompter

Figure 5 presents a formatting example in Policy Prompter.

## D  Experimental Setup

### Datasets.

- Multiwoz 2.0 (Budzianowski et al., 2018) is a **multi-domain** task-oriented dataset, which

| Model | Task | Schema types | Training strategy | | |
|---|---|---|---|---|---|
| | | | Fine-tuning | Pre-training | Prompting |
| RobustSF (Shah et al., 2019) | SF | slot names/value examples | ✓ | | |
| TRADE (Wu et al., 2019) | DST | slot names/value examples | ✓ | | |
| ZSTL-SD (Campagna et al., 2020) | DST | ontology, dialog templates | ✓ (+synthesized data) | | |
| S-DST (Rastogi et al., 2020b) | DST | slot names/descriptions +service, intent names/descriptions | ✓ | | |
| T5DST (Lin et al., 2021b) | DST | slot names/descriptions | ✓ | | |
| TransferQA (Lin et al., 2021a) | DST | slot names/value examples | | ✓ (QA tasks) | |
| IC-DST (Hu et al., 2022) | DST | slot names/value examples | | | ✓ |
| SDM-DST (Wang et al., 2022a) | DST | slot names/value examples | ✓ | | |
| Bert+S (Mosig et al., 2020) | E2E policy | system-side policy skeletons | ✓ | | |
| SAM (Mehri and Eskenazi, 2021) | E2E policy | user-aware policy skeletons | ✓ | | |
| ZSDG (Zhao and Eskenazi, 2018) | E2E dialog | ontology, response templates | ✓ | | |
| DAML (Qian and Yu, 2019) | E2E dialog | ontology, response templates | ✓ | | |
| AnyTOD (Zhao et al., 2022) | E2E dialog | policy programs +slot names/value examples +slot descriptions +user action names/states/descriptions | ✓ | ✓ (heterogeneous tasks) | |
| IG-TOD (Hudecek and Dusek, 2023) | E2E dialog | slot names +slot descriptions | | | ✓ |
| SGP-TOD (ours) | E2E dialog | user-aware policy skeletons (+slot names/value examples) | | | ✓ |

Table 7: Zero-shot task-oriented dialog modeling. (Schema items enclosed in parentheses are required only when accessible.)

contains 8,438/1,000/1,000 dialogs for training/validating/testing, spanning seven domains: restaurant, attraction, train, hotel, taxi, police, and hospital. `Multiwoz 2.0` is annotated with belief states and system actions.

- `Multiwoz 2.2` (Zang et al., 2020) is a improved version of `Multiwoz 2.0`, encompassing refined belief state annotations, slot descriptions, user action annotations, *etc.*

- RADDLE (Peng et al., 2021a,c) consists of four **single-domain** dialog datasets derived from `Multiwoz 2.0` (*i.e.,* restaurant, train, hotel, attraction), reorganized by Peng et al. (2021a). Each corpus contains 50/50/200 dialogs for training/validating/testing, expect for 100 testing dialogs in attraction domain.

- STAR (Mosig et al., 2020) includes 24 tasks in 13 domains (*e.g.,* "apartment" domain comprises "apartment-search" and "apartment-schedule"), requiring the dialog model to conform to the provided task schema. We use 2,688 single-task dialogs from the corpus, which follow a "happy path", *i.e.,* the user is not instructed to execute any action exceeding the schema's expectations. Without additional annotations, STAR only provides a flow chart diagram that outlines the dialog policy for each task. The flow chart outlines the task, including the sequence in which at-

tributes should be asked (for example, ask for the user's name before asking for the hotel name), how to query a database, *etc.*

**Automatic Evaluation Metrics.** We evaluate the end-to-end dialog generation performance using the same metrics as those listed in Budzianowski et al. (2018): (*i*) `Inform`(%) assesses whether the agent returns an acceptable entity. (*ii*) `Success`(%) determines if the agent appropriately responds to each attribute request. (*iii*) `BLEU`(%) (Papineni et al., 2002) measures the word overlap of the generated response against the human response in the corpus. (*iv*) `Combined`(%) judges the overall quality, which is defined as `Combined` = (`Inform` + `Success`) × 0.5 + `BLEU`. Additionally, we utilize `BERTScore`(%) (Zhang* et al., 2020), which focuses on computing semantic similarity between the generated responses and the ground truth, and correlates better with human judgments.

Following Mehri and Eskenazi (2021), we perform the next action prediction task on STAR, which predicts next system action based on the dialog history. Since the system actions and deterministic response templates are mapped one to one in STAR corpus, we believe the end-to-end next action prediction task falls within end-to-end dialog modeling, following Mosig et al. (2020); Mehri and Eskenazi (2021). In addition, we report the results using weighted `F1score`(%) and mean `accuracy`(%).

```
Policy Prompter

Task instruction

  Following the instructions, generate appropriate response based
  on the history.

Formatting example (from other task/domain)

Policy skeleton

(1) user: I'm looking for information on [attraction_name]. action:
attraction (inform (name, address, area, entrance fee)) system:
[attraction_name] is in the [value_area] and their address is
[attraction_address], the entrance fee is [value_count] pounds.
[eos]

(2) user: What is the phone number? action: attraction (inform
(phone), require (more)) system: Their phone number is
[attraction_phone]. is there anything else i can do for you? [eos]

(3) user: I need the area/location and the postcode. action:
attraction (inform (area, postcode)) system: The
[attraction_name] is in the [value_area], their postal code is
[attraction_postcode]. [eos]

(4) user: That is all i need today. thank you! action: attraction
(general (bye)) system: You are welcome , thank you for calling
the [value_place] town info [value_area]. have a great day. [eos]

(5) DB: attraction zero match. action: attraction (inform
(none)) system: There are no results matching your query. Can i
try a different area or type? [eos]

(6) DB: attraction one match. action: attraction (inform (name))
system: We have [attraction_name]. [eos]

(7) DB: attraction five match. action: attraction (inform (choices),
request (area)) system: There are [value_count], which area
were you looking at? [eos]

history

user: Hello, i'm looking for the museum of classical archaeology,
could you tell me more about it? system: Certainly , the
[attraction_name] is located at [attraction_address] and has free
entrance, phone [attraction_phone] . do you also need the
postcode ? user : Yes, i'd like that. thank you so much!

SQL: select * from attraction where name = museum of classical
archaeology DB: Attraction one match

action: attraction (inform (postcode), require (more))
system : The postcode for the [attraction_name] is
[attraction_postcode] . Is there anything else i can help you with ?

  ...
```

Figure 5: A formatting example in Policy Prompter.

**Human Evaluation Metrics.** We employ inter-active human evaluations to assess the quality of dialog agents, following the evaluation protocol in the DSTC9 Track 1 challenge (Gunasekara et al., 2020). We recruit student helpers to help with evaluations. For each dialog session, student helpers are provided with a goal and accompanying instructions, subsequently necessitating a discourse with the agent to achieve the goal via natural language. Upon the conclusion of each dialog session, students are mandated to assess the overall dialog quality employing these five metrics: ($i$) Success w/o g($\%$) evaluates whether the agent accomplishes the task. ($ii$) Success w/ g($\%$) judges whether the agent accomplishes the task and of-

fers matched slot values compared to the database record. ($iii$) Understanding(1-5) quantifies the accuracy with which the agent comprehens user utterances. ($iv$) Appropriateness(1-5) signifies the naturalness, appropriateness and fluency of an agent response. ($v$) Turns denotes the average number of dialog turns within successful dialog sessions.

## E  Implementation Details

### Regarding SGP-TOD:

- LLMs: We employ ChatGPT ("gpt-3.5-turbo"), GPT-3.5 ("text-davinci-003") and Codex ("code-davinci-002") as the fixed LLMs to implement the proposed SGP-TOD. Throughout the evaluation, we set temperature to 0.5.

- DST Prompter – belief instruction: In the context of multi-domain scenarios, the belief instructions encompassing all domains are incorporated, while solely the target domain's belief instruction is introduced in single-domain settings.

- Policy Prompter – policy skeleton: For the Multiwoz datasets, we manually construct the policy skeleton through observing a few dialogs in the training corpus, following Mosig et al. (2020); Mehri and Eskenazi (2021). In the case of the STAR corpus, we employ flow chart diagrams and several dialogs to develop the policy skeleton, following the guidelines set forth by Mehri and Eskenazi (2021). We integrate the relevant user template utterance and the system action into the policy skeleton, thereby augmenting the LLM's understanding of directives, in the absence of belief annotations. The prompt examples for the STAR dataset are shown in Appendix M.

- Formatting example: Following the zero-shot scenario in Wang et al. (2022b), we insert one formatting example from different tasks (fixed through the experimental procedure) into the prompt. The formatting example employed within DST Prompter/Policy Prompter is randomly chosen from the training corpus of different tasks/domains, conforming to zero-shot scenario proposed by Wang et al. (2022b). We appraise multiple randomly selected formatting examples, the evaluation results reveal minor deviations. In the experiments on domain

extension (Section 4.5) and ablation analysis (Section 4.7), we employ the same (two) formatting exemplar turns originating from other domains within the RADDLE corpus for all prompting techniques.

**Regarding compared methods:**

(*i*) **Zero-shot transfer methods:**

- BERT+S (Mosig et al., 2020) is a schema-guided method that augments a BERT-base classifier (Devlin et al., 2019) with a provided system-side schema to predict the next system action.

- SAM (Mehri and Eskenazi, 2021) represents a schema-guided model based on BERT-base, which aligns the dialog context to a user-aware schema to predict the next system action.

- ANYTOD-XXL (Zhao et al., 2022) adopts a neural LM to track dialog states and user actions utilizing slot and action descriptions. Then a program that outlines a predefined task policy is executed to recommend appropriate system actions. Upon considering these system actions, an LM generates the ultimate system action and formulates the corresponding template response using the approach proposed by Kale and Rastogi (2020). ANYTOD-XXL is implemented on T5-XXL (Roberts et al., 2022) and pre-trained on SGD dataset (Rastogi et al., 2020a)[8]

(*ii*) **Prompting methods:**

- IG-TOD-CHATGPT (Hudecek and Dusek, 2023) is a prompting approach based on ChatGPT that leverages the dialog context and manually-crafted slot descriptions as the prompt, to track dialog states, fetch DB entries, and produce responses. IG-TOD-CHATGPT-ZS and IG-TOD-CHATGPT-FS are in the zero-shot and few-shot settings, respectively.

- FEW-SHOT-CHATGPT is a few-shot prompting strategy implemented on ChatGPT, where we use a few (*i.e.,* $k$) dialog turns, randomly sampled from the training corpus to instruct ChatGPT on task execution. Upon evaluating

various configurations of $k$, the optimal results manifest with 15 on Multiwoz (2.0 and 2.2), and 10 on RADDLE, exhibiting no further substantial enhancements.

- SGP-TOD (Ours) is a schema-guided prompting strategy, which is compatible with any off-the-shelf LLMs. In this paper, we employ ChatGPT ("gpt-3.5-turbo"), GPT-3.5 ("text-davinci-003") and Codex ("code-davinci-002") as the fixed LLMs. Following the zero-shot scenario in Wang et al. (2022b), we insert one formatting example from different tasks (fixed through the experimental procedure) into the prompt. More implementation details are provided in Appendix E.

## F   Zero-Shot End-to-End Evaluation Results on STAR

Figure 6 exhibits the zero-shot evaluation results on STAR, utilizing varying amounts of training dialogs (ranging from 1 to 1,000) and formatting example turns (spanning from 1 to 10) from source domains/tasks. SGP-TOD, *merely with two formatting sample turns*, achieves superior or comparable performance compared to BERT+S, SAM, which are fine-tuned on adequate source data.

We observe that SGP-TOD, *employing only two formatting sample turns*, attains superior or commensurate performance in terms of both F1score and Accuracy, when compared to SAM trained with 1,000 dialogs. Given that a single dialog contains more than 10 dialog turns, this result suggests that SGP-TOD diminishes labeling expenses by a minimum factor of 1,000. Furthermore, it is noteworthy that augmenting the quantity of formatting exemplar turns exerts a negligible influence on the performance of SGP-TOD.

## G   Ablation Study

Table 8 exhibits the findings from an ablation investigation, addressing the effects of the three integral aspects of SGP-TOD in conjunction with the database expertise, implemented on Multiwoz 2.0 and 2.2, employing GPT-3.5.[9] Combining the three elements in SGP-TOD with the database expertise produces optimal results across both datasets. The removal of the Policy Prompter, database knowledge, and DST Prompter leads to consistent declines in all evaluation metrics, underscoring the

---

[8]The Schema-Guided Dialog (SGD) dataset constitutes a comprehensive, large-scale, multi-domain corpus encompassing over 16,000 dialogs that span across 16 distinct domains.

[9]We inject the same two formatting example turns into the prompt throughout the evaluation.

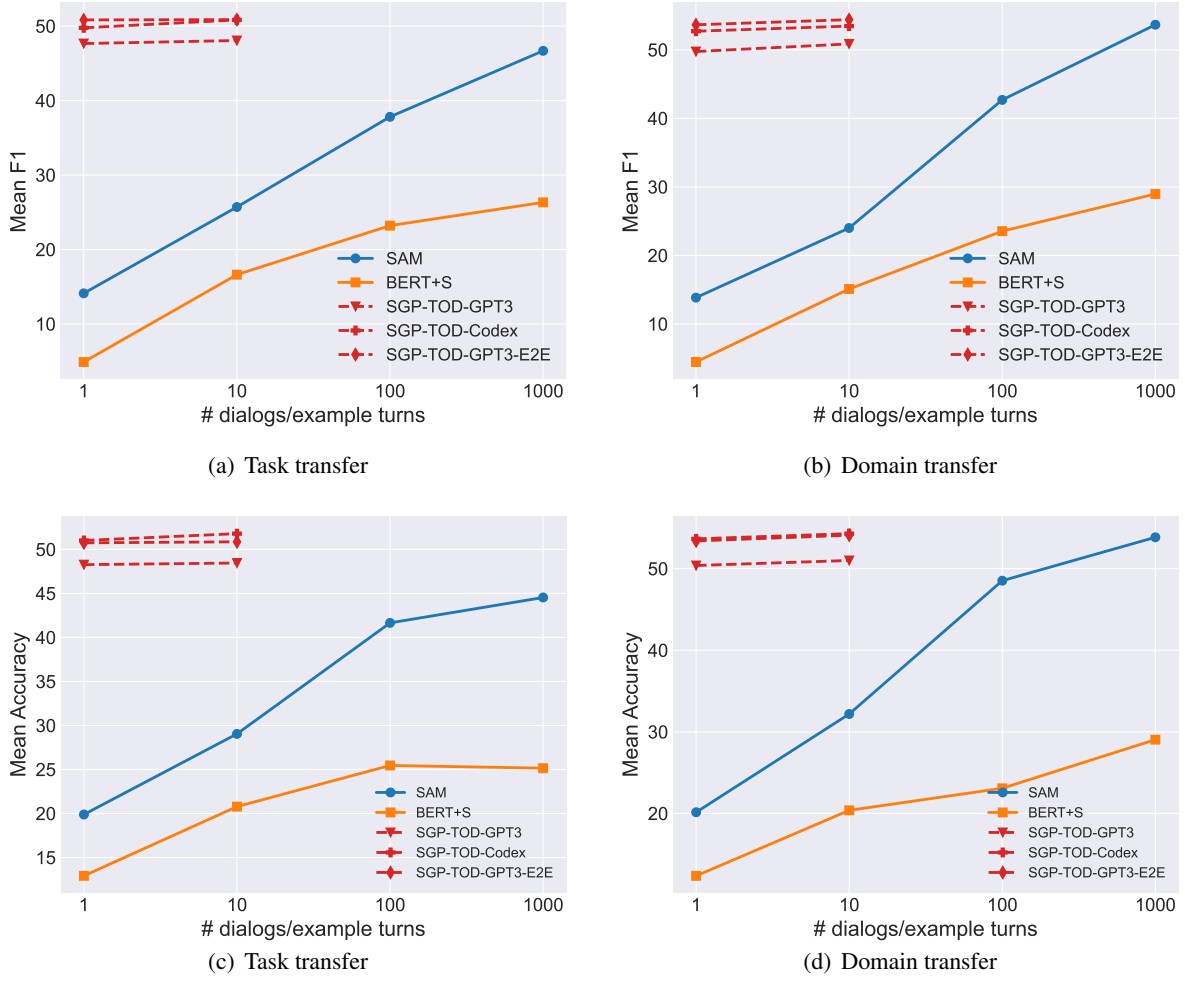

Figure 6: Zero-shot end-to-end evaluation results on STAR with different numbers of training dialogs (1, 10, 100, 1,000) / demonstration example turns (1, 10) from source domain/tasks.

value of enhancing the fixed LLM with the task schema and external database information.

Specifically, **GPT-3.5** (in the final row) exhibits commendable zero-shot performance, highlighting the need of exploiting its superior zero-shot generalization capabilities in dialog generation tasks. Additionally, **Disabling the Policy Prompter** incurs a discernible decline in performance regarding Success (approximately $16\%$) and BLEU (roughly $3\%$), as the Policy Prompter's primary function is to provide task completion guidelines and interaction patterns. **Eliminating the database expertise** primarily reduces Success (by approximately $4\%$), implying that incorporating database information contributes to task completion. Lastly, **excising the DST Prompter** engenders a considerable diminution in performance concerning Inform (around $43\%$) and Success (nearly $18\%$), due to the DST Prompter's intended purpose of assisting the frozen LLM in apprehending the dialog context.

## H   Human Evaluation Details

We enlisted 5 student helpers (*i.e.,* undergraduate students possessing basic proficiency in English communication) to participate in the evaluations. For each dialog agent, we collected 50 dialogs for analysis. Followed the methodology proposed by Li et al. (2022), we generated user goals through the subsequent techniques: (*i*) Randomly selecting slots and slot values within the Restaurant domain from RADDLE corpus to construct a user goal; (*ii*) Replacing the slot values of the user goals in randomly chosen dialogs from the Restaurant corpus with corresponding new values from randomly sampled database entries, thus forming a new user goal; (*iii*) Merging the user goals of several randomly selected dialogs from the Restaurant corpus to create a composite user goal. Lastly, we randomly chose 50 distinct user goals from these newly generated goals.

| Model | Multiwoz 2.0 | | | | Multiwoz 2.2 | | | |
|---|---|---|---|---|---|---|---|---|
| | Inform | Success | BLEU | Combined | Inform | Success | BLEU | Combined |
| SP-TOD-GPT3.5 | **83.88** | **69.87** | **9.09** | **85.97** | **82.00** | **72.50** | **9.22** | **86.47** |
| -policy | 82.28 | 55.65 | 6.51 | 75.48 | 81.80 | 56.20 | 6.63 | 75.63 |
| -policy -DB | 81.20 | 50.95 | 6.48 | 72.56 | 81.40 | 52.30 | 6.57 | 73.42 |
| -policy -DB -belief | 38.74 | 33.13 | 6.18 | 42.12 | 38.60 | 33.90 | 6.29 | 42.54 |

Table 8: Ablation study on the impact of the three components in the proposed SGP-TOD and the database expertise on `Multiwoz` using GPT-3.5. -policy: removing Policy Prompter, -DB: removing database information, -belief: removing DST Prompter.

## I Human Evaluation Results

Figure 7 shows the interactive human evaluation results. SGP-TOD-CHATGPT exhibits a more stable performance. In contrast to the automated evaluation results shown in Table 2, FEW-SHOT-CHATGPT significantly outperforms SOLOIST over all metrics. This indicates that corpus-based evaluations might be biased, given that real user inputs tend to be more dynamic, complex, even with noise. Notably, SGP-TOD-CHATGPT consistently excels compared to the other methods in both evaluations, implying its robustness in handling diverse user inputs.

## J More Details and Results on Domain Extension

**Setup.** Following Zhang et al. (2022), we construct the `Restaurant-ext` corpus by extending the pre-existing `Restaurant` in RADDLE (Peng et al., 2021c) with additional functions. Specifically, we introduce four new slots: *[restaurant_dish]*, *[value_price]*, *[start_time]*, and *[end_time]*. The initial slot pertains to recommendations for signature restaurant meals, while the final three concern delivery service details. All database entries are updated with corresponding values. Table 10 exhibits a dialog example on domain extension. The associated `Restaurant-Ext` database entry is illustrated in Table 9.

**Compared Methods.**

- ChatGPT, GPT-3.5 denote zero-shot prompting with base LLMs that receive merely two formatting example turns from other domains in RADDLE.[10]

- SGP-TOD-CHATGPT, SGP-TOD-GPT3.5 represent our SGP-TOD implementation, with the `Restaurant` policy skeleton.

- SOLOIST is trained with 50 training dialogs in the `Restaurant` domain (previously reported in Table 2).

- SOLOIST+TEACH is fine-tuning method enhanced with machine teaching (Simard et al., 2017). Machine teaching is an efficient approach to equip deployed task bots with the ability to handle new functions by correcting representative failed human-bot dialogs. We deploy SOLOIST to converse with real users, then implement machine teaching via Conversational learner (Shukla et al., 2020), an effective machine teaching tool, to obtain 10/50/50 examples in `Restaurant-ext` for training, validating, and testing. Finally, we fine-tune SOLOIST with gathered 10 training dialogs covering four new slots, resulting in dialog agent SOLOIST+TEACH.

- FEW-SHOT-GPT3.5+TEACH is the few-shot prompting strategy augmented with machine teaching. Based on GPT-3.5, we utilize 10 randomly selected dialog turns from the collected 10 training dialogs as the prompt (with peak performance at 10), resulting in FEW-SHOT-GPT3.5+TEACH.

- SGP-TOD-CHATGPT-EXT, SGP-TOD-GP3.5-EXT refer to SGP-TOD with `Restaurant-Ext` policy skeleton, where we only add four template turns about four new slots to the policy skeleton of `Restaurant`.

**Results.** *Comparison with Base LLMs.* The substantial improvement of SGP-TOD-CHATGPT-EXT and SGP-TOD-GPT3.5-EXT over ChatGPT and GPT-3.5 illustrates SGP-TOD's efficiency in supplying task-specific knowledge in a zero-shot way.

*Impact of Different LLMs.* SGP-TOD-CHATGPT-EXT attains a lower BLEU yet a comparable BERTScore, suggesting that ChatGPT generates more diverse responses.

---

[10]We utilize the same formatting example turns in all zero-shot prompting methods.

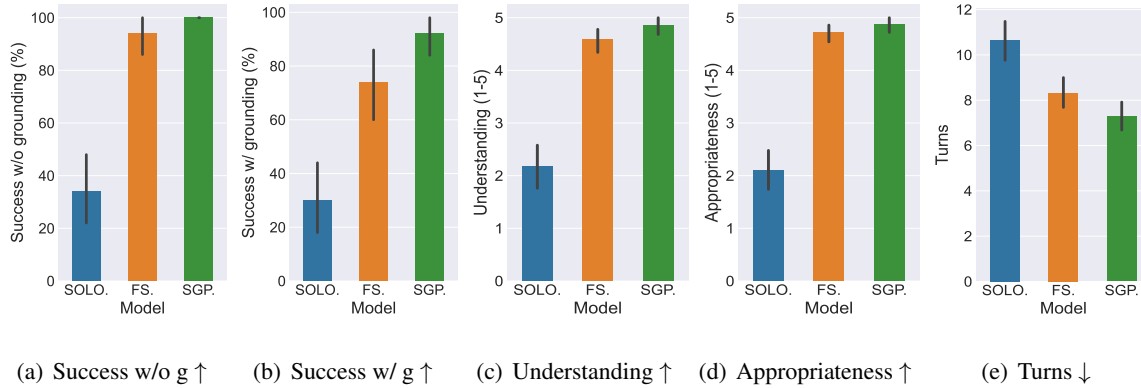

(a) Success w/o g ↑     (b) Success w/ g ↑     (c) Understanding ↑     (d) Appropriateness ↑     (e) Turns ↓

Figure 7: Interactive human evaluation results. SOLO.: SOLOIST, FS.: FEW-SHOT-CHATGPT, SGP.: SGP-TOD-CHATGPT.

| Slot | Value |
|------|-------|
| "address" | "21 - 24 Northampton Street" |
| "area" | "west" |
| "food" | "british" |
| "id" | "14810" |
| "location" | [52.21031, 0.11381] |
| "name" | "saint johns chop house" |
| "phone" | "01223353110" |
| "postcode" | "cb30ad" |
| "pricerange" | "moderate" |
| "type" | "restaurant" |
| "delivery" | "yes" |
| "delivery fee" | "6 pounds" |
| "dish" | "Beef Wellington" |
| "start_time" | "10:30 am" |
| "end_time" | "22:40 pm" |

Table 9: An example of `Restaurant-Ext` DB entry. The newly introduced slot-value pairs relevant to the extended functionality are highlighted.

## K  Case Study

Despite the superior performance of the proposed SGP-TOD on GPT-3.5, we showcase interactive examples utilizing ChatGPT, a renowned and potent chatbot. In Table 11, a user engages with ChatGPT (left) and SGP-TOD-CHATGPT-EXT (right) to complete the identical task on `Restaurant-Ext`.[11] The user initiates the conversation by seeking recommendations for a Tuscan restaurant with no price range preference. Lacking external database information, ChatGPT conveys inaccurate details (Turn 2), whereas SGP-TOD-CHATGPT-EXT informs users of the absence of matching restaurants and proposes alternatives (Turn 2). This exemplifies the benefits of integrating real-world expertise into the fixed LLM. Furthermore, ChatGPT persistently inquires about

the desired price range despite the user's indifference. We argue that SGP-TOD assists the frozen LLM in discerning user intentions. In Turn 4, ChatGPT continues to furnish fabricated details (*i.e.,* the restaurant name and phone number) concerning the nonexistent eatery, while SGP-TOD-CHATGPT-EXT identifies a suitable Korean restaurant and the corresponding factual information. In contrast with ChatGPT, SGP-TOD-CHATGPT-EXT adeptly addresses inquiries about the delivery service (Turn 6), indicating that SGP-TOD is capable of endowing the frozen LLM with novel functionalities.

## L  Our Proposed SGP-TOD with a Dialog Example

Figure 8 illustrates the proposed SGP-TOD with a dialog example.

## M  Prompt Examples for STAR Dataset

Two exemplars of prompts for `STAR` dataset are presented in Figure 9 and Figure 10.

---

[11]ChatGPT and SGP-TOD-CHATGPT-EXT are previously reported in Table 4. The same two formatting example turns are incorporated into the prompt for both zero-shot strategies.

| Turn | Utterance |
|---|---|
| 1 | **User:** I want a restaurant on the *west* end of town. |
| 2 | **System:** Ok. Are you looking for any particular type of food? |
| 3 | **User:** I *don't care*. |
| 4 | **System:** Are you looking for a particular price range? |
| 5 | **User:** A *moderately priced* restaurant would be good. |
| 6 | **System:** How about Saint Johns Chop House? They serve moderately priced British food on the west side of town. |
| 7 | **User:** Can i please have their address and phone number? |
| 8 | **System:** Saint Johns Chop House's address is 21-24 Northampton Street, their phone number is 01223353110. |
| 9 | **User:** Does the restaurant offers delivery service? If it does, what's the delivery fee? |
| 10 | **System:** Yes, it does, and the delivery charges 6 pounds. |
| 11 | **User:** Thank you, goodbye. |
| 12 | **System:** Thank you for using our system. Goodbye. |

Table 10: A dialog example on domain extension. Task bots need learn to provide pertinent responses concerning the expanded delivery service in subsequent interaction turns, adapting to the continually evolving user/enterprise demands.

| Turn | Interaction between a user and ChatGPT | Turn | Interaction between a user and SP-TOD-ChatGPT-Ext |
|---|---|---|---|
| 1 | **User:** Hi, i'm looking for a place with *Tuscan* food in *any* price range. | 1 | **User:** Hi, i'm looking for a place with *Tuscan* food in *any* price range. |
| 2 | **System:** Certainly, there are 3 *Tuscan* restaurants in town. What is your preferred price range? | 2 | **System:** I am sorry, there are no *Tuscan* restaurants. Here are some other options: *Korean, Italian, Chinese*. Which one would you like more information on? |
| 3 | **User:** I *don't care* about the price. Please recommend one for me, i also need the phone number please. | 3 | **User:** How about any *Korean* restaurants? I also need the phone number please. |
| 4 | **System:** The phone number for *Pizza Hut City Centre* is *01223323737*. Is there anything else i can help you with today? | 4 | **System:** *Little Seoul* is a *Korean* restaurant. Their phone number is *01223308681*. |
| 5 | **User:** Does the restaurant offer delivery service? How much does the delivery charge? | 5 | **User:** Does the restaurant offer delivery service? How much does the delivery charge? |
| 6 | **System:** I'm sorry, but I could not find any information on delivery fee in our database. You may want to contact the restaurant directly to inquire about this. Is there anything else I can assist you with today? | 6 | **System:** Yes, they offer delivery service and the delivery charge is *4 pounds*. Can I help you with anything else? |
| 7 | **User:** No. Thank you, goodbye. | 7 | **User:** No. Thank you, goodbye. |
| 8 | **System:** Glad i could assist you. Have a good day. | 8 | **System:** You're welcome, have a great meal! Goodbye! |

Table 11: Illustration of two interactive examples between a user and CHATGPT (left), SGP-TOD-CHATGPT-EXT (right). Appropriate responses and inappropriate responses are highlighted.

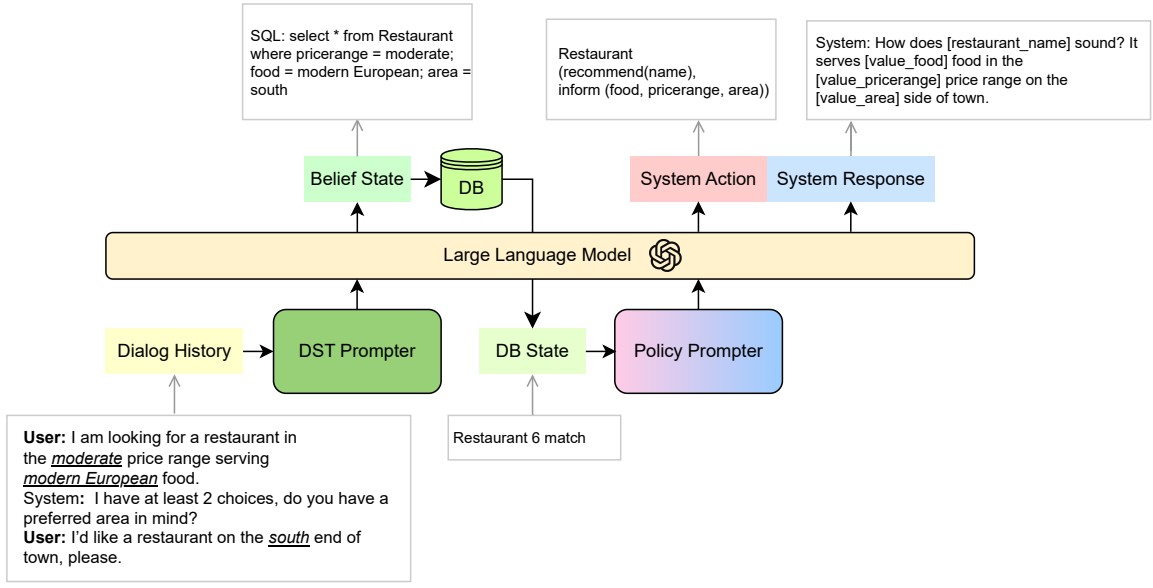

Figure 8: Illustration of the proposed SGP-TOD with a dialog example. Note that the belief state in the represented in the SQL format, the details of which are described in Section 3.3.

**Policy Prompter**

**Task instruction**

Generate appropriate system actions based on the history, following the most relevant task rule.

**Formatting example (from other task/domain)**

**Task schema**

(1) **user:** Hello [sep] **system action:** hello [eos]

(2) **user:** I'd like to find out the status of my ride [sep] **system action:** ask_name [eos]

(3) **user:** My name is [NAME] [sep] **system action:** ride_ask_booking_number [eos]

......

**History**

user: Hi, i need to check my ride status.

**Answer**

(2) **user:** i'd like to find out the status of my ride [sep] **system action:** ask_name [eos]

**Test (on target task/domain)**

**Task schema**

(1) **user:** hello [sep] **system action:** hello [eos]

(2) **user:** what is the weather like? [sep] **system action:** weather_ask_day [eos]

(3) **user:** on [DAY_OF_WEEK] [sep] **system action:** weather_ask_location [eos]

......

**History**

user: i want to know the weather for Tuesday.

**LLM**

Answer

(3) user: on [DAY_OF_WEEK] [sep] system action: weather_ask_location [eos]

Figure 9: Policy Prompter of SGP-TOD on STAR. The relevant template turn within the input, the generated user template utterance, and the system action in the output are accentuated.

**Policy Prompter**

**Task instruction**

Generate appropriate system actions based on the history, following the most relevant task rule.

**Formatting example (from other task/domain)**

**Task schema**

(1) **user:** Hello [sep] **system action:** hello **system:** Hello, how can i help? [eos]

(2) **user:** I'd like to find out the status of my ride [sep] **system action:** ask_name **system:** Could you give me your name, please? [eos]

(3) **user:** My name is [NAME] [sep] **system action:** ride_ask_booking_number **system:** Can i get your booking ID, please? [eos]

......

**History**

user: Hi, i need to check my ride status.

**Answer**

(2) **user:** i'd like to find out the status of my ride [sep] **system action:** ask_name **system:** Could you give me your name, please? [eos]

**Test (on target task/domain)**

**Task schema**

(1) **user:** hello [sep] **system action:** hello **system:** Hello, how can i help? [eos]

(2) **user:** what is the weather like? [sep] **system action:** weather_ask_day **system:** For what day would you like the weather forecast? [eos]

(3) **user:** on [DAY_OF_WEEK] [sep] **system action:** weather_ask_location **system:** For what location would you like the weather forecast? [eos]

......

**History**

user: i want to know the weather for Tuesday.

**LLM**

Answer

(3) user: on [DAY_OF_WEEK] [sep] system action: weather_ask_location system: For what location would you like the weather forecast? [eos]

Figure 10: Policy Prompter of SGP-TOD-E2E on STAR. The relevant template turn in the input, the generated user template utterance, the system action and the system response in the output are highlighted.