# OpenReview forum: "SGP-TOD: Building Task Bots Effortlessly via Schema-Guided LLM Prompting"
_EMNLP/2023/Conference — EMNLP 2023 Findings_

### Official Review · Reviewer_msez · 2023-07-27

**Soundness:** 2

**Excitement:**

3: Ambivalent: It has merits (e.g., it reports state-of-the-art results, the idea is nice), but there are key weaknesses (e.g., it describes incremental work), and it can significantly benefit from another round of revision. However, I won't object to accepting it if my co-reviewers champion it.

**Paper Topic And Main Contributions:**

This paper proposes schema-guided prompting to build task-oriented dialog systems (TOD) with large language models (LLM) in zero-shot settings. It introduces a Dialog State Tracking Prompter to aid LLM to track dialog state explicitly and a Policy Prompter to direct the generation of responses. Experimental results show strong zero-shot performance on various datasets.

**Reasons To Accept:**

1. The method has strong performance on various benchmark datasets.
2. This paper demonstrates how previous techniques for TOD like dialog state tracking can effectively help design prompts for LLM-based chatbots.

**Reasons To Reject:**

1. Analyses are not sufficient to support the effectiveness of the two prompters. (1) Tab. 6 suggests that the improvement is mainly from the DST prompter. (2) It is unclear whether the two complex prompter is more effective than simply introducing meta-instructions to let the LLM track states and make policies.


**Reproducibility:**

3: Could reproduce the results with some difficulty. The settings of parameters are underspecified or subjectively determined; the training/evaluation data are not widely available.

**Reviewer Confidence:**

4: Quite sure. I tried to check the important points carefully. It's unlikely, though conceivable, that I missed something that should affect my ratings.

---

> ### Author Rebuttal · Authors · 2023-08-29
>
> **Reply-to-Reasons-To-Reject:**
>
> **"1) Effectiveness of the two prompters."**
>
> - In the submitted manuscript, we reported our main results in Table 1 and explicated them in Section 4.2. The proposed SGP-TOD-ChatGPT notably surpasses IG-TOD-ChatGPT-zs (Hudecek and Dusek, 2023) with respect to the Success score (54.70\% vs 20.00\%), a state-of-the-art prompting technique employing slot descriptions for tracking dialog states, fetching DB entries, and generating responses (detailed in Section 4.1). This underscores the indispensability of our proposed policy prompter in SGP-TOD for guiding LLMs towards generating appropriate responses for new tasks. To answer this specific question, we also conducted an experiment on SGP-TOD-ChatGPT by excluding the Policy Prompter and report the result in below table. The result indicates that, with only the DST prompter, SGP-TOD-ChatGPT-w/o Policy still substantially surpasses IG-TOD-ChatGPT-zs in terms of Success (41.60\% vs 15.00\%). This superior performance additionally substantiates the efficacy of the proposed DST prompter.
> - In Table 6, the DST Prompter significantly contributes to the performance of SGP-TOD-GPT3.5 (approximately 31\% increase over Combined Score), which demonstrates the efficacy of the proposed DST Prompter. In addition, the Policy prompter further advances the Combined score by nearly 11\%. This notable advancement is remarkable given the intricate nature of the well-established Multiwoz 2.2 dataset (Zang et al., 2020), the challenging research objective of zero-shot end-to-end dialog generation, and the robust baseline exemplified by GPT3.5 (the last row in Table 6).
>
> We would greatly appreciate it if you could provide more detailed comments.
>
> |            Model           | Multiwoz 2.2 |         |      |          |
> |:--------------------------:|:------------:|---------|------|----------|
> |                            | Inform       | Success | BLEU | Combined |
> | IG-TOD-ChatGPT-zs          | -            | 15.00   | 3.58 | -        |
> | SGP-TOD-ChatGPT-w/o Policy | 68.80        | 41.60   | 3.75 | 58.95    |
>
> **"2) Comparison to meta-instructions."**
>
> - Thank you for your insightful comments. Exploring meta-instructions to empower LLMs for decision-making could lead to a highly intriguing direction. The concept of the proposed prompters essentially represents a form of meta-instruction. Through experiments, we have demonstrated that our approach outperforms existing prompting methods for Task-Oriented Dialogue. Nevertheless, we do acknowledge that there might exist other uncomplicated and easily comprehensible formats for LLMs to track states and generate actions.  We will leave it as future work.
>
> **Reply-to-Reproducibility:**
>
> - Parameter settings: Comprehensive experimental details have been provided in Appendix E.
>
> - Training/evaluation data: All utilized datasets, which are well-studied and publicly accessible, have their details delineated in Appendix D.
>
> - Prompter examples pertaining to our proposed approach can be found in Appendices C and M. Furthermore, as stated in the abstract, we intend to release our code and data upon acceptance.

---

### Official Review · Reviewer_t1ZP · 2023-08-04

**Soundness:** 4

**Excitement:**

4: Strong: This paper deepens the understanding of some phenomenon or lowers the barriers to an existing research direction.

**Paper Topic And Main Contributions:**

The paper describes an LLM prompting-based approach to task oriented dialogue. Using exemplars and prompts for the state tracking, dialogue policy, and response generation tasks, the model uses LLM prompting to perform all those tasks in an E2E fashion. Results are shown on MultiWoZ 2.0 and 2.2, and also the RADDLE and STAR benchmarks - both the regular variants of these datasets as well as the domain extension setting on STAR.

**Questions For The Authors:**

- It may be useful to discuss parameter counts when comparing results. Even though prompting and fine-tuning are vastly different learning mechanisms, the models being used differ greatly in size from some of the baselines, which makes at least a mention of parameter counts worthwhile.

**Reasons To Accept:**

- The paper presents a sound, simple, flexible prompting-based technique to take up all TOD sub-tasks in E2E fashion
- Results look fairly strong, comparing favorably with nearly all the finetuned baselines. The non-finetuning baselines compared against are all quite recent and strong ones as well.
- The paper is clearly written and the modeling approach is explained in sufficient detail.

**Reasons To Reject:**

- Human evaluation results compare only to SOLOIST, which is a relatively old paper at this point. It is unclear just how convincing the human evaluation is, for that reason.

**Reproducibility:**

4: Could mostly reproduce the results, but there may be some variation because of sample variance or minor variations in their interpretation of the protocol or method.

**Reviewer Confidence:**

4: Quite sure. I tried to check the important points carefully. It's unlikely, though conceivable, that I missed something that should affect my ratings.

---

> ### Author Rebuttal · Authors · 2023-08-29
>
> **Reply-to-Reasons-To-Reject: "Human evaluation results."**
>
> - Thank you for the comment. Our primary focus of this paper is on zero-shot and few-shot prompting evaluation. While we acknowledge that SOLOIST is somewhat dated, it remains a prominent model for MultiWoZ dataset. Furthermore, human evaluation results have shown that SGP-TOD significantly outperforms Few-shot-ChatGPT and SOLOIST, with success rates surpassing them by 18 and 62 points, respectively (Table 5). These findings align with automatic evaluation results and show the effectiveness of Schema-Guided Prompting.
>
> **Reply-to-Q1: "Parameter counts."**
>
> - Thank you for the valuable suggestion! We will add the details on parameter counts  in the next version.

---

### Official Review · Reviewer_5kY2 · 2023-08-05

**Soundness:** 4

**Excitement:**

4: Strong: This paper deepens the understanding of some phenomenon or lowers the barriers to an existing research direction.

**Paper Topic And Main Contributions:**

Goal: To leverage LLMs for developing task-oriented dialog (TOD) systems in zero-shot setting.

Main Idea: Prompt LLM with task schema and dialog context to generate DST and policy outputs.

Details
1. The paper proposes SGP-TOD that prompts LLMs using task-specific ontology (possible slot-value pairs) and flow outlines.
2. SGP-TOD has two prompters. DST Prompter queries LLM for DST in SQL format for a given dialog history. Policy Prompter then utilizes flow outlines, DST, DB state and dialog history and queries LLM for action and response.
3. Authors benchmark SGP-TOD on Multiwoz 2.0/2.2, RADDLE and STAR datasets and show that SGP-TOD has significant gains against few-shot methods and sets new zero-shot SOTA.
4. Authors also perform human evaluation across models which is a good addition.


**Questions For The Authors:**

1. How are the dialog flows in Policy Prompter different from few-shot exemplars? Is exclusion of generated SQL query the only difference? Can Policy Prompter be truly called zero-shot?
2. How are the dialog flow outlines selected?

**Reasons To Accept:**

1. SGP-TOD demonstrates how to systematically leverage LLMs for TOD task. Based on the benchmarking, SGP-TOD (GPT3.5) achieve SOTA zero-shot performance.
2. Prompt design in SGP-TOD can be relevant for other tasks with complex reasoning.

**Reasons To Reject:**

SGP-TOD uses dialog flow guidelines which are partial dialog context with gold delexicalized responses. It is unclear how these dialog flows are different from the few-shot setting. This raises a question on SGP-TOD to be a zero-shot model.

**Reproducibility:**

4: Could mostly reproduce the results, but there may be some variation because of sample variance or minor variations in their interpretation of the protocol or method.

**Reviewer Confidence:**

3: Pretty sure, but there's a chance I missed something. Although I have a good feel for this area in general, I did not carefully check the paper's details, e.g., the math, experimental design, or novelty.

---

> ### Author Rebuttal · Authors · 2023-08-29
>
> **Reply-to-Reasons-To-Reject and Q1:**
>
> Thank you for your questions. We answer your questions from the following two perspectives:
>
> **"1) What is the difference between dialog flows and few-shot exemplars?"**
>
> - *Dialog flows* are abstract representations of dialogue policies. They explicitly describe anticipated interaction patterns (Peng et al., 2021b) and are typically depicted using graphs (such as the flowcharts in the STAR corpus (Mosig et al., 2020). However, in the *few-shot setting*, the few-shot exemplars consist of examples sampled from the training data. These exemplars encode very limited interaction patterns, illustrating partial dialogue policies.
>
> **"2) Why is SGP-TOD a zero-shot model?"**
>
> - As illustrated in Figures 3 and 5 and explained in Section 3.4, the policy prompter takes dialog flows as input, which sketch the dialog policies. Each element within these dialog flows is textualized, appearing as a delexicalized user utterance/database state, a system action, and a system response. Since the policy prompter takes *zero* examples from the dataset as input, we note it as a *zero-shot prompting* method, following Mosig et al. (2020) and Mehri and Eskenazi (2021).
>
> **Reply-to-Q1:**
>
> - Please see the detailed response provided above, under “Reply-to-Reasons-To-Reject and Q1”.
>
> **Reply-to-Q2: “Selection of dialog flow outlines.”**
>
> - Dialog flows that describe the functions of dialog systems in an abstractive way are typically pre-built when developing task-oriented dialogs systems.  For the particular datasets used in this paper, similar to Mosig et al. (2020) and Mehri and Eskenazi (2021), we manually create the outline by examining the task descriptions, provided by the dataset. More details are provided in Appendix E.

---

### Meta-Review · Area_Chair_aLog · 2023-09-25

**Recommendation:** 3

**Metareview:**

The paper benchmarks prompting for zero-shot TOD using DST and Policy prompt mechanisms. Using ChatGPT, the paper demonstrates SOTA performance on zero-shot TOD using schema-guided prompting.

This is primarily a benchmarking paper and does not introduce any core technical innovation. However, reviewers appreciated the overall prompting framework. There were some concerns around complexity of the overall prompting setup, towards which, the authors clarified on the necessities of the modules in certain scenarios.

The paper argues schema-guidance to be better than few-shot prompting. As such, discussing about the paper - "Show, Don’t Tell: Demonstrations Outperform Descriptions for Schema-Guided Task-Oriented Dialogue" might be useful.

---

### Decision · Program_Chairs · 2023-10-07

**Decision:**

Accept-Findings

**Comment:**

The paper benchmarks prompting for zero-shot TOD using DST and Policy prompt mechanisms. Using ChatGPT, the paper demonstrates SOTA performance on zero-shot TOD using schema-guided prompting.

This is primarily a benchmarking paper and does not introduce any core technical innovation. However, reviewers appreciated the overall prompting framework. There were some concerns around complexity of the overall prompting setup, towards which, the authors clarified on the necessities of the modules in certain scenarios.

The paper argues schema-guidance to be better than few-shot prompting. As such, discussing about the paper - "Show, Don’t Tell: Demonstrations Outperform Descriptions for Schema-Guided Task-Oriented Dialogue" might be useful.